# Swollen Feet: Considering the Paradoxical Roles of Interleukins in Nephrotic Syndrome

**DOI:** 10.3390/biomedicines12040738

**Published:** 2024-03-26

**Authors:** Maria E. Kovalik, Monique A. Dacanay, Steven D. Crowley, Gentzon Hall

**Affiliations:** 1Division of Nephrology, Duke University, Durham, NC 27701, USA; ginakovalik@gmail.com (M.E.K.);; 2Duke Molecular Physiology Institute, Duke University, Durham, NC 27710, USA

**Keywords:** nephrotic syndrome, podocytes, interleukins, cytoprotection, immunotherapeutics

## Abstract

Interleukins are a family of 40 bioactive peptides that act through cell surface receptors to induce a variety of intracellular responses. While interleukins are most commonly associated with destructive, pro-inflammatory signaling in cells, some also play a role in promoting cellular resilience and survival. This review will highlight recent evidence of the cytoprotective actions of the interleukin 1 receptor (IL-1R)- and common gamma chain receptor (IL-Rγc)-signaling cytokines in nephrotic syndrome (NS). NS results from the injury or loss of glomerular visceral epithelial cells (i.e., podocytes). Although the causes of podocyte dysfunction vary, it is clear that pro-inflammatory cytokines play a significant role in regulating the propagation, duration and severity of disease. Pro-inflammatory cytokines signaling through IL-1R and IL-Rγc have been shown to exert anti-apoptotic effects in podocytes through the phosphoinositol-3-kinase (PI-3K)/AKT pathway, highlighting the potential utility of IL-1R- and IL-Rγc-signaling interleukins for the treatment of podocytopathy in NS. The paradoxical role of interleukins as drivers and mitigators of podocyte injury is complex and ill-defined. Emerging evidence of the cytoprotective role of some interleukins in NS highlights the urgent need for a nuanced understanding of their pro-survival benefits and reveals their potential as podocyte-sparing therapeutics for NS.

## 1. Introduction

Nephrotic syndrome (NS) is a clinical diagnosis characterized by severe proteinuria (>3.5 g/24 h), hypoalbuminemia (<3 g/dL) and edema [1,2]. NS is the most common glomerular disease in children and adults [3]. The annual incidence of idiopathic NS is estimated to be 7/100,000 in children and 3/100,000 in adults [1,4]. NS is an umbrella diagnosis that encompasses several forms of glomerular disease with multiple histologic phenotypes (e.g., minimal change nephropathy, focal segmental glomerulosclerosis, membranous nephropathy, diabetic nephropathy, etc.). NS results from the injury or loss of glomerular visceral epithelial cells (i.e., podocytes) [5,6]. Podocytes are an essential cellular component of the tripartite glomerular filtration barrier that synthesize and maintain the filtration slit diaphragm [7]. The slit diaphragm is an elaborate, “ladder-like” assembly of proteins that spans the interdigital spaces between adjacent podocyte foot processes [8,9,10]. It is thought that this dynamic macromolecular network functions like a sieve to retain desirable molecules in the blood and allow the passage of waste into the primary urinary filtrate [8,9,10]. NS can occur across the lifespan and may be caused by a variety of insults (e.g., genetic, environmental toxins, medications, viral infection, systemic illness, etc.) [4]. Broadly, NS is classified as steroid-sensitive (SSNS) or steroid-resistant (SRNS) based on the clinical response to corticosteroid treatment [4]. SSNS accounts for 80–90% of pediatric cases. The remaining 10–20% are SRNS and nearly 30% of these are attributable to single-gene defects [11]. SRNS is more common in adults and manifests most frequently with FSGS histology [12]. Classically, idiopathic NS is thought to be a T-cell-mediated disorder that is incited and/or driven by a circulating permeability factor in the blood [13,14,15,16,17]; however, our understanding has evolved to include a prominent role for B-cells in driving the disease [18,19]. While the identity of the circulating factor(s) remains elusive, several have been proposed, including immune mediators such as tumor necrosis factor-α [20,21], CD40 [22] and cardiotrophin-like cytokine-1 (CLC-1 or CLCF-1), a member of the interleukin-6 family [23]. Adding to the complexity, we and others have observed that the disease onset and severity are also influenced by the cell types expressing these pro-inflammatory mediators [24]. These findings and many others underscore the importance of immune mediators in the pathogenesis of NS and justify the pursuit of a more nuanced understanding of their influence on podocyte health and disease.

This review will focus on the known roles of interleukins in NS, highlighting recent evidence of the paradoxical roles of the IL-1R- and IL-Rγc-signaling cytokines in podocyte cytoprotection. A comprehensive discussion of the roles of interleukins in kidney disease is provided by Mertowska et al. [25].

## 2. Deleterious Effects of Interleukins in NS: IL-1R Superfamily

The IL-1 superfamily is comprised of 11 members (IL-1α, IL-1β, IL-1ra, IL-18, IL-33, IL-37, IL-36α, IL-36β, IL-36γ, IL-36ra and IL-38) that signal through isoforms of the IL-1 receptor [26]. IL-1α and IL-1β are the most widely known members of the family, most commonly associated with pro-inflammatory, cytodestructive signaling in virtually all tissues of the body [26]. IL-1 family cytokines are broadly classified as pyrogens, owing to their ability to induce fever; however, they have also been shown to increase pain sensitivity (hyperalgesia), vasodilation and the relaxation of vascular tone (hypotension) and cell death (e.g., apoptosis, pyroptosis) [26]. IL-1 is widely recognized for its direct and indirect roles in the initiation and propagation of various kidney diseases [25]. Podocytes have been demonstrated to be the primary producers of IL-1α and IL-1β in humans and experimental models of glomerulonephritis [27,28]. Nuclear factor-κB signaling is a principal driver of intrinsic pro-inflammatory signaling in podocytes via the upregulation of IL-1β expression [29,30,31]. Brähler et al. demonstrated that IL-1β stimulation directly induces dysregulated cytoskeletal rearrangement and dysmotility in podocytes and aggravates glomerulosclerosis via NF-κB signaling [30,31]. Additionally, IL-1β has been demonstrated to synergize with the apoplipoprotein-L1 renal risk variants to facilitate the entry and persistence of HIV-1 infection in podocytes [32]. In diabetic nephropathy, IL-1 receptor-associated kinase 1 (IRAK1), a critical downstream effector of IL-1/IL-1R signaling, is upregulated in glomeruli and podocytes and the knockdown of IRAK1 ameliorates renal injury and dysfunction, reduces podocyte apoptosis, increases nephrin expression and preserves podocyte cytoskeletal architecture and glomerular basement membrane integrity [33]. Similar results have been reported with the use of AS2444697, a selective small-molecule inhibitor of IL-1 signaling [34,35]. Notably, the use of highly selective biologic inhibitors of IL-1/IL-1R signaling (i.e., anakinra and canakinumab) have improved glomerular function in humans and experimental models of diabetic nephropathy [36,37]. Interleukin 1 inhibitors have also been shown to reduce proteinuria and preserve glomerular function in a cohort of patients with Familial Mediterranean Fever [38,39], demonstrating the utility of IL-1/IL-1R antagonists for the treatment of certain forms of podocytopathy. Further support for the deleterious role of IL-1R signaling in kidney disease was recently provided by Cho et al. [40]. Using Mendelian randomization analyses to examine the causality between serum interleukin levels and kidney function in the CKDGen and UK Biobank databases, Cho et al. found that higher serum levels of the naturally occurring IL-1R antagonist (IL-1ra) were significantly associated with higher eGFR values [40]. IL-1ra is a 16–18-kDa member of the IL-1 family that binds to IL-1 receptors without activating signaling [41,42]. IL-1ra competitively inhibits the binding of both IL-1α and IL-1β to IL-1R and it is estimated that a 100-fold or greater excess of the peptide over IL-1 may be necessary to inhibit biological responses [42]. These findings support a cytoprotective role for IL-1ra in these kidney disease cohorts.

IL-18 is another IL-1R family member that may contribute to podocyte dysfunction in NS [43]. IL-18 has been shown to be upregulated in the podocytes of patients with ANCA-associated vasculitis [44]. Sugiyama et al. demonstrated that the deletion of the IL-18 receptor ameliorates renal injury in bovine serum albumin-induced glomerulonephritis [45]. IL-18 has also been shown to synergize with IL-12 in minimal change nephropathy to drive the production of vascular permeability factor (VPF/VEGF-A) [46,47,48]. Along with IL-1β, IL-18 is a principal mediator of pyroptotic signaling in podocytes [49,50,51,52], and therapies that reduce signaling through this axis are protective in podocytes [53,54,55,56].

## 3. Deleterious Effects of Interleukins in NS: IL-Rγc Family

The IL-Rγc-family is comprised of six members (IL-2, IL-4, IL-7, IL-9, IL-15 and IL-21), unified by the requirement for the common gamma chain receptor subunit (IL-Rγc) for signaling. While all members of the family require IL-Rγc, only IL-2 and IL-15 require the IL-Rβ subunit as well [57,58]. IL-2 is the most widely known member of the IL-Rγc-family. IL-2 has been shown to be upregulated in idiopathic nephrotic syndrome [59,60,61,62]. IL-2 is a potent inducer of T-cell proliferation and differentiation, supporting its potential role as a key immune modulator of NS [60,62,63]. However, IL-2 has also been shown to injure podocytes directly. Zea et al. have shown that podocytes express the IL-2Rα receptor and that the stimulation of podocytes with IL-2 increased the expression of pro-apoptotic markers, decreased autophagic flux, induced mitochondrial depolarization and activated JAK3/STAT5a signaling. Additionally, IL-2 has been shown to induce the production of vascular permeability factor in minimal change nephropathy [48,64].

IL-4 has been shown to be upregulated in children with minimal change nephropathy [59,65,66]. B-cell-derived IL-4 has been shown to induce proteinuria and foot process effacement in podocytes [67]. Specifically, Kim et al. showed that the overexpression of IL-4 in mice was sufficient to induce kidney injury and proteinuria and that cultured murine podocytes treated with IL-4 exhibited membrane ruffling and widespread foot process retraction [67]. These findings were consistent with those of Lee et al., who observed significant disruption of the podocyte actin cytoskeleton with IL-4 treatment [68].

IL-7 is a member of the IL-Rγc-signaling family that has been considered as a putative circulating permeability factor. Kanai et al. reported elevated serum levels of IL-7 in a small cohort of patients with SSNS [69], and Agrawal et al. suggested that plasma profiling could differentiate children with SSNS from those with SRNS at disease onset based on serum IL-7 levels [70]. Despite these indications of a role for IL-7 in the pathogenesis of NS, few studies have offered mechanistic insights. One study by Zhai et al. showed that IL-7 directly inhibited podocyte nephrin expression, induced apoptosis, caused podocyte actin cytoskeletal reorganization and disrupted transwell filtration barrier function in a rodent model of NS [71].

## 4. Deleterious Effects of Interleukins in NS: IL-6 Family

Interleukin 6 is a pleiotropic cytokine, exerting both pro- and anti-inflammatory effects in tissues [72]. IL-6 is the most extensively characterized member of the family and has been primarily defined as a mediator of pro-inflammatory signaling in the kidney [73]. IL-6 is expressed by glomerular and tubular cells [73], and bacterial lipopolysaccharide (LPS) has been demonstrated to induce the robust expression of the cytokine in cultured podocytes [74]. IL-6 has been shown to induce podocyte hypertrophy in an experimental model of diabetic nephropathy by Jo et al. [75] and myosin light chain hyperphosphorylation, focal contact disassembly and hypermotility in cultured podocytes [76]. Notably, Nagayama et al. showed that the direct treatment of normal mice with IL-6 did not induce glomerular injury, proteinuria or a change in podocyte marker expression relative to untreated controls [77]. Such contradictory findings have led many to question whether IL-6 is an independent pathogenic contributor to podocyte injury in NS, while its role as a driver of mesangial [73], endothelial [78,79] and tubular injury [80,81] in kidney disease is firmly established.

## 5. Deleterious Effects of Interleukins in NS: IL-13

Interleukin 13 has also been shown to play a role in the pathogenesis of NS. In 2007, Lai et al. demonstrated that the overexpression of IL-13 induced a minimal change nephropathy-like phenotype in rats [82]. Specifically, they found that IL-13 overexpression induced minimal change histology, albuminuria and podocyte injury as evidenced by significantly reduced nephrin, podocin and dystroglycan expression [82]. IL-13 also increased the expression of B7-1 in this model [82]. Similar findings were reported by Ha et al. when they demonstrated that IL-13 significantly downregulated and altered the distribution of zonula occludens-1 (ZO-1), synaptopodin, α-actinin, CD2AP and p130Cas in cultured human podocytes [83]. They also showed that IL-13 exposure upregulated β-catenin and B7-1/CD80 expression [83]. B7-1 has previously been established as a mediator of LPS-induced podocyte injury in humans and experimental models of glomerular disease [84,85,86]. The role of B7-1 in minimal change nephropathy and FSGS has been controversial [87]; nonetheless, Ha and others have demonstrated the amelioration of podocyte injury and proteinuria with therapies that reduce B7-1 expression [82,84,88]. In addition to these effects, IL-13 overexpression has also been shown drive the inappropriate activation of the Vav1-Rac1 pathway, which caused pathologic rearrangements of the podocyte actin cytoskeleton in an experimental model of minimal change nephropathy [89].

## 6. Deleterious Effects of Interleukins in NS: IL-17

IL-17 has also been shown to cause podocyte injury in many forms of NS (i.e., minimal change nephropathy, mesangioproliferative glomerulonephritis and FSGS) [90,91]. In 2013, Wang et al. showed that IL-17 was significantly increased in a pediatric NS cohort and that IL-17 induced the downregulation of podocalyxin and apoptosis in cultured murine podocytes [91]. Zhai et al. also showed that IL-17 expression was upregulated in biopsies of patients with primary nephrotic syndrome and correlated positively with podocyturia [90]. Additionally, they showed that IL-17 reduced the expression of the podocyte markers (i.e., Wilms’ Tumor 1, nephrin, synaptopodin and podocalyxin) and increased apoptosis [90]. Similarly, Yan et al. showed that IL-17 induced glomerular injury, proteinuria and podocyte dedifferentiation in a rat model of adriamycin nephropathy [92], and Zhang et al. demonstrated that the administration of IL-17 neutralizing antibodies reduced glomerular injury, preserved podocyte numbers and ameliorated proteinuria in an experimental model of diabetic nephropathy [93].

## 7. Deleterious Effects of Interleukins in NS: IL-20

IL-20 is a member of the IL-10 family. Members of the IL-10 family are recognized as immunomodulatory cytokines that dampen immune responses in the context of disease [94]. Nonetheless, IL-20 has been identified as a driver of podocyte injury in diabetic nephropathy [95]. Hsu et al. showed that IL-20 and its receptor were upregulated in diabetic nephropathy and that IL-20 induced p38, JNK and ERK MAPK signaling, apoptosis and fibrogenic gene expression in podocytes [95], demonstrating an atypical role for this family of cytokines.

“Good and evil are so close as to be chained together in the soul.” Robert Louis Stevenson, Dr. Jekyll and Mr. Hyde

## 8. Cytoprotective Effects of Interleukins in NS: IL-1R Superfamily

The paradoxical roles of the interleukins as mediators of injury and cytoprotection are best exemplified in the actions of the IL-1 superfamily. Despite the aforementioned injurious effects of IL-1R-family cytokines, members of this family have been shown to bolster podocyte resilience and survival. In 1997, Neimir et al. reported, for the first time, that podocytes were the major source of IL-1α and IL-1β in human glomerulonephritis [27]. The authors noted the early and robust expression of IL-1α and IL-1β in diseased glomeruli that waned with the loss of podocyte maturity markers [27]. Based on these findings, they hypothesized that the recruitment of metabolic pathways downstream of IL-1/IL-1R may be elicited to mitigate injury prior to overt podocyte loss [27]. Consistent with this hypothesis, we have shown that IL-1R colocalizes with synaptopodin in podocyte foot processes and with nephrin in the slit membrane to some extent (Figure 1). Additionally, we and others have demonstrated the cytoprotective effects of IL-1/IL-1R signaling in podocytes. For instance, Wright and Beresford demonstrated that podocytes challenged with IL-1β exhibited the marked rearrangement of the actin cytoskeleton [96]. This alteration was transient, preceded by an increase in intracellular calcium and associated with foot process effacement without an increase in apoptosis [96]. They hypothesized that this rearrangement of the actin cytoskeleton may be an early adaptation to prevent apoptosis [96]. Consistent with this hypothesis, IL-1R has been shown to directly associate with RhoA GTPase, a critical mediator of actin cytoskeletal dynamics and anti-apoptotic signaling [97,98,99]. In 2022, Ren et al. showed that IL-1/IL-1R signaling protected podocytes from adriamycin- and nephrotoxic serum-induced nephritis [100]. Specifically, we showed that the podocyte-specific deletion of IL-1R in mice impaired podocyte maturity marker expression, increased albuminuria, increased podocyte apoptosis and decreased AKT activation in response to adriamycin and nephrotoxic serum challenge [100]. The IL-1/IL-1R-induced activation of AKT may be mediated via inhibitor κB kinases (i.e., IKKα or IKKε), which have been shown to mediate pro-survival signaling through the PI-3K/AKT and JAK/STAT pathways [101,102,103]. Notably, we also demonstrated the colocalization of IL-1R with the slit diaphragm protein nephrin [100]. Subsequently, by super-resolution microscopy, we confirmed this observation, showing that IL-1R colocalizes with synaptopodin throughout podocyte foot processes and at their bases near nephrin in the slit membrane (Figure 1). Hall et al. and others have previously demonstrated the assembly of a pro-survival signaling hub at the slit diaphragm composed of proteins such as anillin, nephrin, podocin and CD2AP, which drive cytoprotective signaling through the phosphatidylinositol-3-kinase (PI-3K)/AKT pathway [104,105,106,107]. The localization of IL-1R in foot processes near nephrin may suggest that IL-1/IL-1R contributes to this AKT-mediated cytoprotective signaling module (Figure 1).

In addition to IL-1, IL-37 has also been shown to exert cytoprotective benefits for podocytes in diabetic nephropathy [108]. Zhang et al. demonstrated that IL-37 significantly reduced inflammation, oxidative stress and apoptosis induced by high-glucose challenge [108]. Similar anti-inflammatory benefits were observed with the IL-33 treatment of experimental models of minimal change nephropathy [109]. Specifically, Lui et al. demonstrated that rats treated with adriamycin and puromycin aminonucleoside to induce experimental minimal change nephropathy developed podocyte cytoskeletal disruption, apoptosis and proteinuria [109]. These effects were ameliorated by treatment with IL-33, demonstrating the cytoprotective signaling characteristics of this IL-2 family cytokine [109]. Although IL-33 is most commonly associated with pro-inflammatory signaling, it can also activate protective signaling through the myeloid differentiation primary response 88 (MyD88)/TNF receptor-associated factor 6 (TRAF6)/interleukin 1 receptor-associated kinase (IRAK)/nuclear factor kappa B (NF-κB), MyD88/TRAF6/receptor-interacting protein kinase (RIP)/Akt and MAPK pathways [110]. Because each of these pathways can be cytoprotective, it is possible that they may contribute to IL-33-induced protective signaling in podocytes [110]. Notably, the expression of the IL-33 receptor complex has been confirmed in podocytes and IL-33 expression has been shown to be upregulated in patients with minimal change nephropathy [111].

## 9. Cytoprotective Effects of Interleukins in NS: IL-Rγc Superfamily

IL-Rγc-signaling cytokines are well recognized for their cytoprotective and cytoproliferative actions, principally mediated through the JAK/STAT, PI-3K and MAPK signaling pathways [57,112]. IL-9 has been shown to protect podocytes from early injury and progressive glomerulosclerosis in adriamycin-induced nephropathy [113,114]. Specifically, Xiong et al. demonstrated that podocytes expressed the IL-9 receptor and that IL-9 deficiency enhanced adriamycin-induced podocyte apoptosis [113,114].

The cytoprotective actions of IL-15 in renal epithelial cells are well described [112,115]. In 2001, Shinozaki et al. demonstrated that IL-15 protected kidney epithelial cells from nephrotoxic serum nephritis [116]. Specifically, they demonstrated that IL-15 was robustly expressed throughout the tubular and glomerular compartments and that this was reduced in the setting of nephrotoxic toxic serum nephritis [116]. They showed that IL-15 deficiency enhanced adriamycin-induced renal epithelial cell apoptosis and that this effect could be mitigated with recombinant IL-15 [116]. Further, they demonstrated that the IL-15-mediated cytoprotection is T-cell-independent and that IL-15 acts in an autocrine fashion to prevent renal epithelial cell apoptosis [116]. Similar findings were subsequently reported by Mooslechner et al., who showed that the administration of low-dose IL-15 reduced glomerular injury and albuminuria in mice [117]. More recently, Niasse et al. demonstrated a protective role for IL-15 signaling in podocytes via the activation of the JAK/STAT5B pathway [118]. They showed that IL-15 activated STAT5B expression in podocytes, modulated autophagy, ameliorated glomerular injury and reduced albuminuria in adriamycin-treated mice [118]. Consistent with these findings, we have demonstrated IL-15Rα colocalization with nephrin in healthy human glomeruli (Figure 2) and immortalized human podocytes and showed that *IL-15Rα* deficiency reduces PI-3K/AKT signaling and increases podocyte apoptosis. IL-15/IL-15R signaling may also be of importance in podocyte aging [119]. Telomere shortening has emerged as a potential driver of podocyte senescence in age-associated glomerulopathy [120,121]. IL-15 has been shown to upregulate telomerase activity and expression and prevent senescence in lymphocytes [122,123]. These findings have not been replicated in podocytes, but suggest a role for IL-15 as a preemptive, podocyte-sparing therapy for age-associated glomerulopathy. Despite these benefits, it must be noted that IL-15 has been shown to induce the expression of interferon gamma (INFγ) [124]. INFγ is a potent inducer of apolipoprotein L1 (APOL1) [125] and Nystrom et al. demonstrated that IL-15 modestly induced apolipoprotein 1 (APOL1) expression in glomerular endothelial cells and podocytes [126]. Since it is widely recognized that common pathogenic variants in APOL1 significantly enhance the risk of kidney disease in people of recent African descent [127], this has obvious implications for APOL1-mediated kidney disease (AMKD); however, it is unclear whether IL-15 acts as a direct driver of APOL1 expression, whether it acts indirectly to drive expression through the upregulation of INFγ or whether IL-15 contributes significantly to the pathogenesis of AMKD. Nonetheless, these findings point to a potential therapeutic role for recombinant human IL-15 or IL-15 superagonists in the treatment of podocytopathy. The emergence of IL-15 immunotherapies for the treatment of cancer provides a range of therapeutic options for potential repurposing in NS [128,129]. In addition to recombinant human IL-15, nearly ten IL-15 agonists have been successfully developed and clinically studied [130]. IL-15 immunotherapies display a range of potencies, half-lives and modes of action, which may be of value across the spectrum of NS [129]. The value of IL-15 immunotherapies in cancer is primarily related to their ability to expand lymphocyte populations [129]. In NS, this effect may be undesirable as the disease is thought to arise, in part, from the dysregulation of the immune system [13,14,15,16,17]. In 2022, Zhang et al. observed that excessive IL-15 promotes cytotoxic T-cell-mediated renal injury and lupus nephritis [131]. The findings of Mooslechner et al., which showed the improvement of nephropathy with low-dose IL-15 treatment, may be instructive in defining the range over which recombinant IL-15 and IL-15 superagonists may be beneficial [117]. It is encouraging that the use of recombinant IL-15 and the supergonist ALT-803 (N-803) has been well tolerated in cancer trials, which may bode well for their use in future NS trials [132,133].

## 10. Conclusions

Our understanding of the paradoxical role of interleukins in kidney disease is evolving, but it is clear that the course of disease can be influenced by a complex milieu of influences (e.g., cell type-specific cytokine expression, tissue-specific receptor densities, tissue microenvironments, etc.). While it is clear that many interleukins play a pathogenic role in podocyte injury, emerging evidence of a cytoprotective role for some interleukins suggests that the repurposing of novel interleukin immunotherapeutics may be a new frontier for the treatment of NS (Figure 3). For example, the expanding repertoire of well-tolerated IL-15 agonists/superagonists may provide new opportunities to bolster podocyte resilience and survival. Certainly, recent discoveries highlight the potential for novel interleukin immunotherapies for the treatment of NS and warrant further investigation.

## Figures and Tables

**Figure 1 biomedicines-12-00738-f001:**
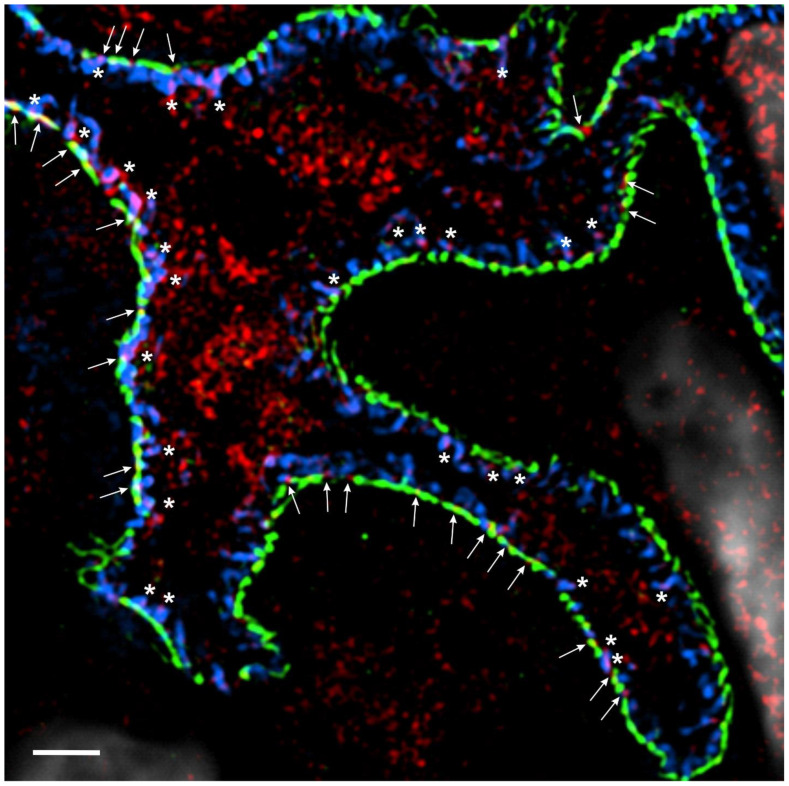
IL-1R is expressed in podocytes. High-resolution microscopy of healthy human kidney sections (3 µm) stained for the slit diaphragm protein nephrin (Progen, cat: GP-N2, 15 μg/mL, green), the actin-associated protein synaptopodin (Progen, cat# 65194, 0.67 μg/mL, blue), IL-1Rα (Thermo Fisher, cat#: PA5-13428, 40 μg/mL, red) and DNA with 4′,6-diamidino-2-phenylindole (DAPI; 1:100) (white). IL-15Rα puncta colocalize with synaptopodin in podocyte foot processes (white asterisks) and with nephrin along the slit diaphragm to some extent (white arrows). Three-dimensional structured illumination microscopy (3D-SIM) images were acquired using an N-SIM super-resolution microscope (Nikon, Melville, NY, USA) with a 100× silicone objective. The images were stitched using NIS-Elements AR (Nikon).

**Figure 2 biomedicines-12-00738-f002:**
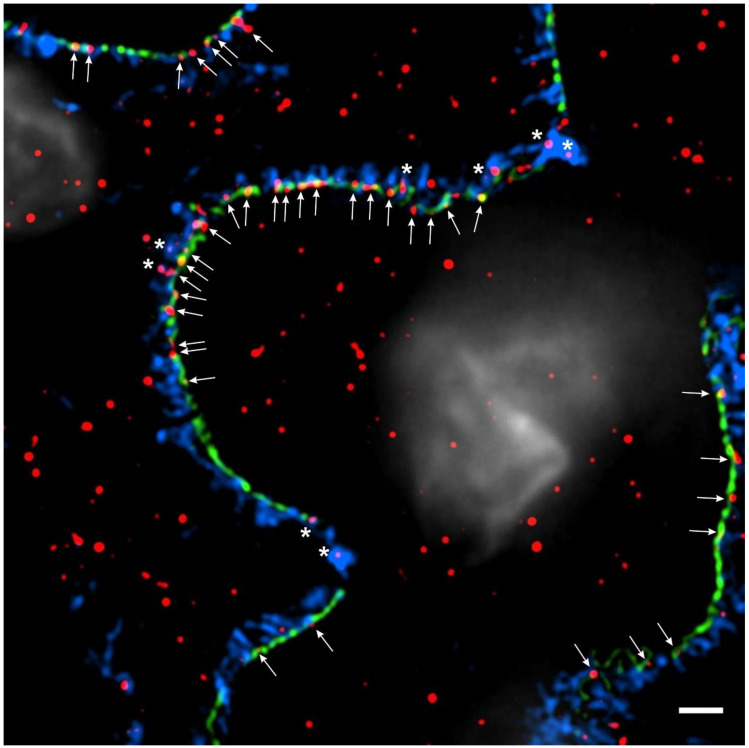
IL-15Rα is expressed in podocytes. High-resolution microscopy of healthy human kidney sections (3 µm) stained for the slit diaphragm protein nephrin (Progen, cat#: GP-N2, 15 μg/mL, green), the actin-associated protein synaptopodin (Progen, cat#: 65194, 0.67 μg/mL, blue), IL-15Rα (R&D Systems, cat#: AF247, 40 μg/mL, red) and DNA with 4′,6-diamidino-2-phenylindole (DAPI; 1:100) (white). IL-15Rα puncta colocalize with synaptopodin in podocyte foot processes (white asterisks) and with nephrin along the slit diaphragm to some extent (white arrows). Three-dimensional structured illumination microscopy (3D-SIM) images were acquired using an N-SIM super-resolution microscope (Nikon) with a 100× silicone objective. The images were stitched using NIS-Elements AR (Nikon).

**Figure 3 biomedicines-12-00738-f003:**
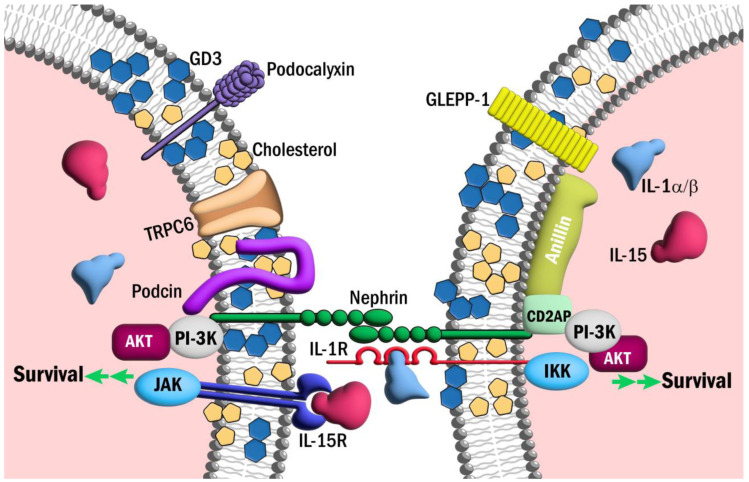
IL-1R and IL-15Ra may contribute to survival signaling in podocyte foot processes. Schematic representation of IL-1R and IL-15Rα in podocyte foot processes. Pathways downstream of these receptors may contribute to AKT-mediated pro-survival signaling in podocytes.

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
