# Peer review of "Swollen Feet: Considering the Paradoxical Roles of Interleukins in Nephrotic Syndrome"

_biomedicines, 2024, doi:10.3390/biomedicines12040738_

Round 1
Reviewer 1 Report
Comments and Suggestions for Authors
This is an interesting review summarizing findings on the role of interleukins, particularly IL-1R and IL-Rgc signaling, in the nephrotic syndrome. This review may provide many clinicians involved in the management of nephrotic patients with clues to the mechanisms of the nephrotic syndrome and provide a methodology for treatment. However, for those actually involved in the clinical management of nephrotic syndrome, it is a somewhat confusing to understand. I have the following concerns;
1. Although the authors present nephrotic syndrome as a T cell-mediated disease in the introduction, recent evidence indicates that idiopathic NS is rather a B cell-mediated disease. In particular, the advent of rituximab, an anti-CD20 monoclonal antibody, has dramatically changed the concept of the pathogenesis of nephrotic syndrome. That is why the authors state "classically," but in this review, I do not see an association between IL-1R and IL-Rgc signaling and B cells or the therapeutic effect of anti-CD20 antibodies. A more robust explanation of these associations would be helpful.
2. In the same context, although the authors introduced that SRNS is more common in adults and manifests most frequently with FSGS histology, but nothing described about membranous nephropathy (MN), a most common cause of idiopathic nephropathy in adult. MN is caused by deposition of immune complexes in the subepithelial region of the glomerular basement membrane, and several target antigens have been identified, including PLA2R and THSD7A in both primary and secondary forms. The relationship between these antigen-antibody responses seen in MN and IL-1R and IL-Rgc signaling is also unknown, similarly may be well worth mentioning in this review.
3. The Methods section begins with a description of immunohistological staining, but it is unclear what was stained with this technique and antibody. The material and its source of origin should be clearly stated.
From this review, I found that each of the many IL-1 families has been characterized on the basis of previous reports and that each is complexly involved in the pathogenesis of nephrotic syndrome, with some acting in an impairing manner on podocytes and others in a protective manner. However, as I commented above, I found it rather difficult to understand their relationship to therapeutic efficacy and true therapeutic targets in nephrotic syndrome. Before concluding, it would be good to include a discussion by the authors based on my earlier comments and why the true targets have not been discovered so far.
Author Response
Dear Colleague,
Thank you for your thorough and thoughtful review of our manuscript. Per our invitation, we shaped this review to highlight recent advances in our understanding of the pathomechanisms podocyte injury/dysfunction in nephrotic syndrome. We focused on the role of interleukins as there have been exciting developments in the past 5 years that shift our understanding of the role of these molecules exclusively as mediators of podocyte injury in glomerular disease. Consequently, this review is not intended to provide an overview of all forms of nephrotic syndrome, only those where these insights have emerged. Please see my responses to your comments/suggestions below:
1.) Thank you for this insight. As mentioned, this review highlights recent insights into the role of interleukins in NS. Per our thorough review of the literature, there have been no insights into the protective role of IL-1 or IL-15 signaling in INS caused by B-cell dysregulation. Certainly, the role of T cells in INS has evolved so as you recommend, I have expanded on my description to include mention of the evolving role of B-cells (lines 44-46).
2.) Thank you again for your comment. As mentioned, this review is intended to highlight recent advancements in our understanding of the role of interleukins in nephrotic syndrome. You are correct in your assessment that advancements in our understanding of the role of interleukins in the pathogenesis of podocyte injury/dysfunction in NS are limited. While there are no studies that address the linkage between various MN epitopes, some evidence correlating levels of IL-35 with disease remission have been reported (PMID: 35983038, 26876383). We have avoided mention of correlation studies in this review if they do not inform mechanism.
3.) I have made the requested adjustments to clarify the methods section (line 308).
Reviewer 2 Report
Comments and Suggestions for Authors
This is a complete an updated review of a topic of interest related to the involvement of various interleukins in the pathogenesis of nephrotic syndrome.
There are some considerations:
1. The authors dedicate a paragraph to detail the nephrotic syndrome. We must consider that nephrotic syndrome is a syndrome, not a specific pathology. This introductory paragraph creates confusion by mixing the syndrome, clinical expression of different primary or secondary nephropathies (minimal change nephropathy, focal segmental glomerulosclerosis, membranous nephropathy, amyloidosis, diabetic nephropathy, among others...) with specific pathologic entities. In fact the last part of this paragraph refers to idiopathic nephrotic syndrome, which mainly includes minimal change disease and focal segmental glomerulosclerosis.
When at the end of the paragraph it says: "This review will focus on the known roles of interleukins in NS..." it should be clear what it refers to, whether to nephrotic syndrome in general, the result of various diseases, or to idiopathic nephrotic syndrome, with a specific pathogenic mechanism.
When the authors talk about NS (nephrotic syndrome) throughout the text, it should be clear what they are referring to.
I recommend rewriting this first paragraph, in order to improve its comprehension and the comprehension of the context of the subsequent text.
I also propose that the title of this first paragraph be changed: instead of "Nephrotic syndrome", it should simply be called "Introduction".
2. This manuscript is a review, so it does not make sense to add a methods section as described. This methodology already appears in the cited articles.
3. Similarly the “Antibodies" section should not be here. I think that the sections "Methods" and "Antibodies" should be removed from the text. If a description of the methodology of Figures 1 and 2 is considered relevant, a link to a separate document should be added as a supplement.
Other minor comments:
- Line 236 and 297: In the text of figures 1 and 2, I believe that the title is grammatically incomplete: “IL-1R is expressed podocytes” and “IL-15Ra is expressed podocytes”
- Line 280 - (AMKD) should appear without parenthesis
- Line 332-333: In Referfences, I think reference 1 is cited incomplete and inaccurately
Author Response
Dear Colleague,
Thank you for your thorough and thoughtful review of our manuscript. We have amended the introduction and its title as you suggested to clarify the references made to nephrotic syndrome. We moved a sentence that appeared midway in the original paragraph describing NS as an umbrella diagnosis to an earlier position in the introduction to clarify the idea that multiple disease processes can result in nephrotic syndrome. We also clarified other references to NS throughout the manuscript as suggested.
All minor comments were addressed as noted. Thank you for your attention to those details. As it relates to the methods and antibodies sections, we incorporated your and reviewer 1's comments and moved the sections to the end of the manuscript.
Round 2
Reviewer 1 Report
Comments and Suggestions for Authors
This revised manuscript makes it easier to understand the very large number of interleukins involved in the pathogenesis of nephrotic syndromes. However, there may still be some confusion, especially for treating clinicians, as to which are the true therapeutic targets. Hopefully this will be clarified in the near future and brushed up by the authors in the next report. I have no further comments.
Author Response
Dear colleague,
Thank you for your comments. We will be sure to keep you astute observations in mind for future opportunities we may have to prepare a manuscript from a more clinically-focused perspective. Many thanks again.
Reviewer 2 Report
Comments and Suggestions for Authors
Dear authors: As I have already mentioned, since this manuscript is a review, it is not necessary to add a methods section, either at the beginning or at the end. Including the "Methods" section and the "Antibodies" table adds confusing information, as it is not clear which studies they refer to.
However, if the author still considers it necessary to include them, they should be added in a comprehensible way: The "Antibodies" table should be added as a table, and the "Methods" section should be included as an appendix at the end and referenced in the text where they provide relevant additional information.
Moreover,
Line 32: NS is not a typical or common presentation of IgA nephropathy
Author Response
Dear Colleague,
Thank you again for your thorough and insightful review of our manuscript. Thank you for catching my mistake with the insertion of IgA Nephropathy. I intended to insert minimal change nephropathy instead. I have made that correction to line 32. Also, I have integrated eliminated the methods and antibodies sections as requested and moved the relevant information to the figure legends. Many thanks again.